



# Observed improvement in air quality in Delhi during 2011-
# 2021: Impact of mitigation measures
**Yesobu Yarragunta[1,2], Latha Radhadevi[1*], Aditi Rathod[1], Siddhartha Singh[3], Murthy**
**Bandaru[1]**
**1 Indian Institute of Tropical Meteorology, Dr. Homi Bhabha Road, NCL PO, Pune, India – 411008**
**2. Earth Science Department, Khalifa University, Abu Dhabi, UAE**
**3 India Meteorological Department, Mausam Bhavan, Lodhi Road, New Delhi,**
**India - 110003**
**\*Corresponding author: latha@tropmet.res.in**



**Abstract**
Assessing long-term air quality trends helps evaluate the effectiveness of adopted air pollution control policies.
A decade of SAFAR observations revealed that the trend of particulate matter ($PM_{2.5}$ and $PM_{10}$) in Delhi shows
a reduction of $2.98 \pm 0.53$ μg/m$^3$/y ($4.91 \pm 1.01$ μg/m$^3$/y) or overall 29% (23.7%) reduction between 2011 and
2021 while vehicles almost doubled but with the implementation of cleaner technologies and stricter industrial
regulation. Seasonal negative trends of pre-monsoon (March-April-May; $-3.43 \pm 1.02$ μg/m$^3$/y) and post-
monsoon (October-November; $-4.51 \pm 1.59$ μg/m$^3$/y) are relatively higher. The role of trends in dust storms, fire
counts and annual rainy days are also discussed. The contribution of meteorology to the trend is estimated using
WRF-Chem simulation of $PM_{2.5}$ for October when maximum stubble burning occurs and gets transported to
Delhi. The model is run with the meteorological initial conditions of 2018, 2015, and 2011 while keeping the
emissions of 2018 with identical model configuration and found that meteorology contributed 9.8% in October,
while the observed decline in $PM_{2.5}$ is 35% (best fit) and 25% (value). The study identifies the governmental
control measures at various levels and green initiatives as the significant contributors to air quality improvement
during 2011-2021.
**Keywords**: Air Quality Index; Policy implementation; Particulate Matter; WRF-Chem; Dust storms; Crop
residue burning



## 1. Introduction

Air pollution has recently been one of India's most severe environmental problems, especially in metropolitan cities like Delhi (Beig et al. 2020; 2021; Chen et al., 2020). The economic liberalization led to India's economy becoming one of the world's fastest-growing economies. During the latter half of the 20th century, fast economic growth, rapid industrialization, increased transportation demand, along with rapid urbanization dramatically increased air pollutant emissions. High levels of particulate matter concentrations affected human health and caused broader concern in recent years (Balakrishnan et al., 2019; Geng et al., 2021). In addition, high concentrations also modulate radiative balance through indirect and direct effects (Seinfeld and Pandis., 2006). Ground-level particulate matter such as $PM_{10}$ and $PM_{2.5}$ are extensive environmental problems in metropolitan cities throughout the world (Zhang et al., 2019; Zhang et al., 2020; Beig et al., 2020; Chen et al., 2020; Chen et al., 2023). Delhi is one of the world's most polluted/populated metropolitan cities (Beig et al., 2019; Jena et al., 2021). Several emission sources of anthropogenic origin in urban areas lead to deterioration of air quality, e.g. combustion of fossil fuel and bio-fuel, industrial, re-suspended dust. A wide range of emissions and meteorology conditions affect these sources, formation, chemical composition and transformation of PM in different regions (Zhao et al., 2013; Shrivastava et al., 2015). $PM_{10}$ and $PM_{2.5}$ are the major pollutants in the world's urban areas; hence, National Ambient Air Quality Standards (NAAQS) have been set up for such pollutants in India, similar to many other countries over the globe. The levels established by the Government of India for $PM_{2.5}$ and $PM_{10}$ are 60 $\mu gm^{-3}$ and 100 $\mu gm^{-3,}$ respectively. These levels are frequently exceeded in Delhi (MoEFCC, 2015).

Several policies have been implemented across Delhi in various emission sectors to curb the rising levels of pollutants. Various measures have been taken in the industrial sector, including relocating/shifting, strict emission standards, restrictions on coal use, and particulate filters. Exhaust emissions have been controlled through a variety of measures, including the formation of strict emission standards, reducing sulfur in diesel fuel, reducing benzene in gasoline, introducing unleaded gasoline, clean fuels, scrapping old vehicles, and improving public transportation (Guttikunda et al., 2014). In addition, biomass burning was banned, an Odd-Even vehicle policy was implemented (2016), the National Air Quality Index was introduced (2016), diesel vehicles older than ten years were deregistered (2016), and a Graded Response Action Plan (GRAP) for Delhi-NCR (2017) was implemented. Badarpur thermal power plant was closed (2018), Bharat Stage BS-VI grade auto fuels were used in Delhi in April 2018, and the National Clean Air Program (NCAP) was launched in 2019 (MOEF & CC, 2019). Therefore, it is essential to assess long-term trends along with the policy implementation timeline to study these policies' impact on significant pollutants.

Regional air quality models have been essential tools for scientifically understanding the distribution of emissions sources, transport and transformation (Yarragunta et al., 2020; Shahid et al., 2021; Jena et al., 2021; Du et al., 2022; Kumar et al., 2022). For regional modelling studies, emission inventories are essential for reflecting the emission inputs into the atmosphere. In addition, meteorological conditions play an essential role in forming ground-level $PM_{2.5}$ and $PM_{10}$, and it is necessary to consider the effects when developing emission control strategies in different regions of India. Recently, machine-learning models have been developed to estimate the concentration of air pollutants, removing the impact of meteorology (Zhang et al., 2020; Du et al., 2022; Chen et al., 2023). These algorithms have an improved performance compared to traditional statistical



and chemistry transport models i.e. Weather Research and Forecasting model coupled with Chemistry, WRF-
Chem. through changing bias/variance and error in high-dimensional data sets. However,  Vu et al., (2019)
found that, it is difficult to interpret the underlying mechanism responsible for such change and interpretation of
results of these models. Therefore, chemical transport models are widely used to evaluate air quality response to
clean air policy. However, the operations of the models consume considerable computing resources, and there
are major uncertainties in emission inventories and the models themselves (Zhang et al., 2019). The uncertainty
problems of chemical transport models are checked by their ability to reproduce observations using the
measured data set, i.e. the measured $PM_{2.5}$ and $PM_{10}$. The studies on the relative contribution of emission control
and meteorology to particulate pollution by machine learning model and chemical transport model are very
sparse in the Indian region but are many over different regions of the world (Wang et al., 2019; Choi et al.,
2019; Zhang et al., 2021; Yin et al., 2021). Recently, Hammer et al., (2021) found that the observed decline in
$PM_{2.5}$ during the COVID-19 lockdown in the North China Plain was driven by a combination of emission
reduction and meteorology. Du et al., (2022) found that changes in meteorological factors and emission
reduction contributed to a decrease in $PM_{2.5}$ by 18.6% and 10.5%, respectively, in the Beijing-Tianjin-Hebei
(BTH) region in 2020 compared to 2018. In another study by Singh et al., (2021), during 2014-19, a significant
decline in PM was found in five Indian mega cities such as New Delhi, Chennai, Hyderabad, Mumbai and
Kolkata, ranging from 2-8% per year. Long-term analysis of criteria pollutants over Delhi showed decreasing
trend during 2015-19 (Verma & Nagendra, 2022).
Despite the measures taken by local authorities, very few studies indicate that air quality in Indian cities is
declining significantly. The relative contribution of emission control and meteorology to the variation in $PM_{2.5}$
and $PM_{10}$ is sparse in the Indian context. Thus, evaluating the impact of meteorological variation on pollutants
during recent years was necessary and could provide crucial information for future air pollution control policies.
In this study, we present an analysis of the linear trends of $PM_{2.5}$ and $PM_{10}$ using observed data, and the factors
driving these trends are analyzed with nested WRF-Chem simulations over Delhi. The relative contribution of
meteorological variation to the change in linear trends of $PM_{2.5}$ and $PM_{10}$ in Delhi from 2011 to 2020 is
investigated. The influence of seasonal external factors like dust storms and stubble burning is quantified. Our
study reveals the impacts of meteorological conditions on $PM_{2.5}$ and $PM_{10}$ concentration during the recent
decade 2011-21, for the first time and provide a reference for formulating future air quality policies. The
observation, model configurations and validation are shown in Section 2. The main results and discussions are
presented in Section 3, and the conclusions are given in Section 4.
**2. Methods and materials**
**2.1 Observational network**
The observational network, SAFAR, 'System of Air Quality and Weather Forecasting and Research (SAFAR)'
was commissioned in Delhi in 2010. This pilot project was adopted by GURME and World Meteorological
Organization (Beig et al., 2015). SAFAR-Delhi comprises a network of 10 online automatic Air Quality
monitoring stations (Table.*S1*) and a coupled high-resolution online chemistry transport model, WRF-Chem
(Marrapu et al., 2014; Srinivas et al., 2016) for Air quality prediction. These air quality monitoring stations
(AQMS) are instrumented with US-EPA approved monitors in continuous monitoring mode, spread across





Delhi over different micro-environments viz. background, residential area, traffic location, downtown area and
so on to represent the local environment and the average can be representative of overall Delhi as per WMO
guidelines (Beig et al., 2015; Srinivas et al., 2016). These analyzers are operated and maintained as per the US-
EPA-approved standard specification, and quality control is certified by Bureau Veritas Certification (ISO9001).
The instruments are calibrated based on the Standard Operating Procedures adopted by US-EPA. Details of
SAFAR network in Delhi can be found in Beig et al., (2020; 2021).
**2.2 Model setup**
The detailed description of the SAFAR air quality forecasting model adopted in this work is provided elsewhere
(Marrapu et al., 2014; Srinivas et al., 2016), hence not discussed in detail. It is based on WRF–Chem (Weather
Research and Forecasting coupled with Chemistry) configured with 4-nested domains. There are a total of 33
vertical model layers, with the model top situated at 50 hPa. The National Centre for Environmental Prediction
(NCEP) final analysis fields (FNL) at a resolution of $1° \times 1°$ were used to provide the model with meteorological
initial and lateral boundary conditions. We took the daily varying BB (Biomass burning) emissions of different
trace species from the Fire Inventory from NCAR (National Centre for Atmospheric Research) (FINN)
(Wiedinmyer et al., 2011). Biogenic emissions of trace species were calculated online using the Model of
Emissions of Gases and Aerosols from Nature (MEGAN) (Guenther et al., 2006). We have used the gas-phase
mechanism of CBMZ chemistry scheme consisting of Carbon Bond Mechanism version Z (CBMZ), which
contains 73 chemical species and 237 reactions, and MOSAIC-4 bin (Model for Simulating Aerosol Interactions
and Chemistry; Zaveri et al., 2008) aerosol scheme that uses four sectional bins where three bins are assigned
for aerosols of diameter less than 2.5 μm, and other bin describing the size range 2.5–10 μm. The various
parameterization schemes, input setting and emission inventory used for this WRF-Chem configuration can be
found in detail elsewhere (Marrapu et al., 2014). The model results were routinely validated with surface
observations over the Delhi region, and results can be found elsewhere (Marrapu et al., 2014; Sahu et al., 2015;
Srinivas et al., 2016; Beig and Sahu, 2018; Beig et al., 2021).
**2.3 Influence of seasonal external factors and meteorological conditions**
**2.3.1 Seasonal external factors (Dust storms and stubble burning)**
Northern India (Delhi and Indo Gangetic plain) witnessed several dust storm episodes in May and June due to
low-level jet streams which brought dust particles from the Middle East and especially from the Thar desert
(Dey et al., 2004; Goel et al., 2020; Sethi et al., 2020). Dust events were identified from observations when the
ratio of $PM_{2.5}$ to $PM_{10}$ was less than or equal to 15%, indicating the predominance of coarse/dust particles. These
dust    events    were    also    corroborated    by    NASA's    dust    score    from    Aqua    satellite
(https://worldview.earthdata.nasa.gov). On this basis, we have estimated the number of dust events and the trend
in the occurrence of dust events over the period 2011-2021.
Stubble/biomass burning (majorly during October-November) in the northwest region (mainly Punjab and
Haryana states) is an external factor that significantly impacts air quality in Delhi (Beig et al., 2020) through
transport. Average radiative power, with 40% and 80% confidence, retrieved from Aqua and Terra satellites
data (https://firms.modaps.eosdis.nasa.gov/)   over   the   potential   stubble-burning   region   were   analysed   to




understand the possible significance of Delhi's air quality trend since Delhi's air quality is mainly dependent on
PM, the trends of PM are considered for the current study.
Variations in rainfall during 2011-2021 could affect $PM_{10}$ and $PM_{2.5}$ concentrations. Hence the trend in the
number of annual rainy days was also analysed using gridded rainfall data.
**2.3.2 Meteorological conditions**
Three simulations with the same emission inventory and changing meteorological conditions were conducted
with the setup described in section 2.2 to examine the effects of meteorological conditions on air quality,
particularly on PM. The simulation period was October 2011, 2015, and 2018. The emission inventory 2018 was
used for all the simulations and is considered a reference year for the assessments. The quantitative impact
assessment method of meteorological conditions on PM ($PM_{2.5}$ and $PM_{10}$) was established as follows:
$$M\_PM_{ij} = \frac{PM_{ij} - PM_{2018j}}{PM_{2018j}} X100 \qquad (1)$$
where $PM_{ij}$ the simulated concentration of pollutant j in $i^{th}$ year; $PM_{2018j}$ is the simulated concentration of
pollutant j in 2018 and the unit is μg/ m³. $M\_PM_{ij}$ is the simulated % contribution of meteorological variation to
pollutant j in $i^{th}$ year compared to 2018. Positive values represent unfavourable meteorological conditions in $i^{th}$
year compared to 2018, such as higher relative humidity and lower wind speed, and negative values represent
favourable meteorological conditions in $i^{th}$ year compared to 2018. The method has been widely used by various
researchers (Zhang et al., 2021; Hammer et al., 2021; Du et al., 2022), while the conclusion from model results
can be affected by simulation bias due to uncertainty in chemical mechanisms, emission inventory and
meteorology parameters (Yin et al., 2021). Any error in the simulated PM due to errors in the emission
inventory used gets cancelled, contributing to meteorology alone, as all other inputs remain the same.
**2.4 Trend estimation**
Analysis of long-term trends of air pollutants has significant implications for identifying the emission hot spots,
evaluating the effectiveness of policies and regulations, assessing the health impacts, and understanding the
chemistry and radiative effects of the atmosphere. We have followed the method for trend analysis used by
various researchers ( Brockwell and Davis, 2002; Solmon et al., 2015; Zhang et al. 2017; Georgoulias et al.
2019; Choo et al. 2020; Singh et al. 2021). The monthly averaged concentrations of $PM_{2.5}$ and $PM_{10}$ are used for
the trend calculation over Delhi for 2011-2021. The monthly datasets are first de-seasonalized by applying a 13-
month moving average for trend first guess and after that, a stable seasonal filter is used to remove the seasonal
cycle. Linear regression is applied on the de-seasonalized time series of $PM_{2.5}$ and $PM_{10}$ to calculate the linear
trend. Statistical significance of the linear trend is calculated using a parametric student t-test and the
statistically significant non-zero slopes (p-value < 0.05) are presented.
**3. Results and discussions**
**3.1 Air Quality Index (AQI)**





An AQI is a rating system that describes how clean the air is and how it affects human health. It provides
information in colour and simple numbers without any units for easy understanding. As per CPCB guidelines,
there are six AQI categories: Good, Satisfactory, Moderate, Poor, Very Poor and Severe. AQI for SAFAR
network cities is calculated based on the criteria pollutants viz, $O_3$, CO, $NO_2$, $PM_{10}$, and $PM_{2.5}$. The computation
of AQI requires the concentration of these pollutants and their breakpoint concentration, and details are
available in MoEFCC (2015).

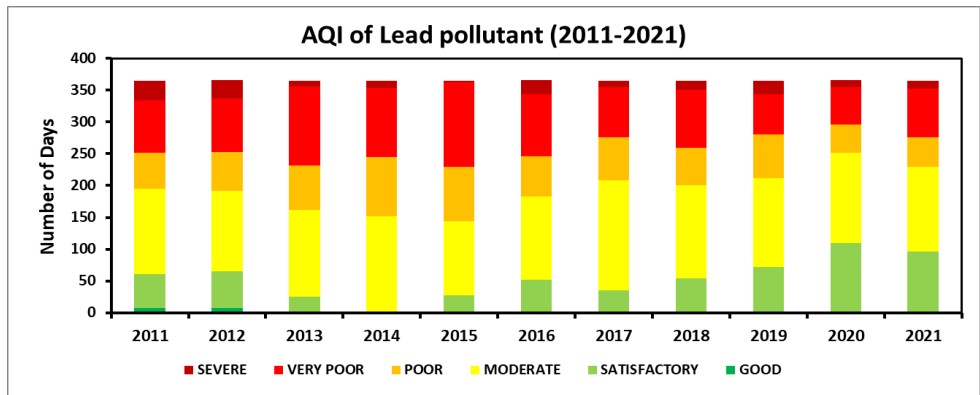

**Fig. 1 Annual variation of AQI over Delhi, 2011-2021**
The annual variation of AQI for lead pollutants over Delhi during the period of 2011-2021 is depicted in Fig. 1.
It represents the number of days that fall into various AQI categories such as Good, Satisfactory, Moderate,
Poor, Very Poor and Severe in each year. AQI in the Moderate category has the highest occurrence in all the
years for the study years. A significant variation is evident in the average number of days falling in each AQI
category. 'Moderate' AQI is reported 32 - 47% of days during the period 2011-21, with an average of 38%,
followed by 'Very Poor' AQI,(25%), 'Poor' (18%) and 'Satisfactory'(14%) while "Severe" was at 4%. More
than 50% of days since 2016 have been in the 'Good to Moderate' AQI category. However, for the year of
Covid lockdown, 2020, 69% (252 days) of days fell into this category, followed by 63 % (229) in 2021, 58%
(212) in 2019 and 57% (208) in 2017. In contrast, the AQI category of severe to poor had increased in the earlier
years from 2011 to 2016. While 'Severe and Poor' combined AQI days were 61 % (221days) in 2015, the same
stood at 31 % (114 days) in 2020. The study results show that days with 'Satisfactory' AQI level have increased
consistently since 2015 while days of 'Very poor' category have decreased, indicating that there has been a
gradual improvement in air quality from 2015 to 2021.
**3.2 Climatology of $PM_{2.5}$ and $PM_{10}$**
Fig. 2a shows the annual average $PM_{2.5}$ and $PM_{10}$ mass concentrations over Delhi during 2011-2021, averaged
across ten stations in different micro-environments. By averaging the data, inhomogeneity can be eradicated,
and the data can be viewed as representative of the entire city area, as explained in section 2.1. The
climatological (2011-2021) average of $PM_{2.5}$ mass concentration was found to be 104±55 μg/m$^3$ with the highest
value of 113 μg/m$^3$, observed in the year 2016 and the lowest one 83 μg/m$^3$ in 2020. Similarly, the average
$PM_{10}$ was found to be 209±85 μg/m$^3$ during this period, with the highest concentration of 229 μg/m$^3$ in 2012



and the lowest 163 µg/m³ in 2020. The linear trends are discussed in the next section for the period 2011-2021
in which 2020 is an anomalous year with full or partial lockdowns implemented during March-May due to the
pandemic. In order to understand whether the trend during 2011-2021 is affected by including 2020 data, we
have calculated anomaly of each year from the decadal mean (2011-2021) and depicted in Figure 2b.The
negative anomalies of PM2.5 in 2019, 2020 and 2021 are almost the same, hence inclusion of 2020 data hardly
changes the annual trend for the period, 2011-2021.

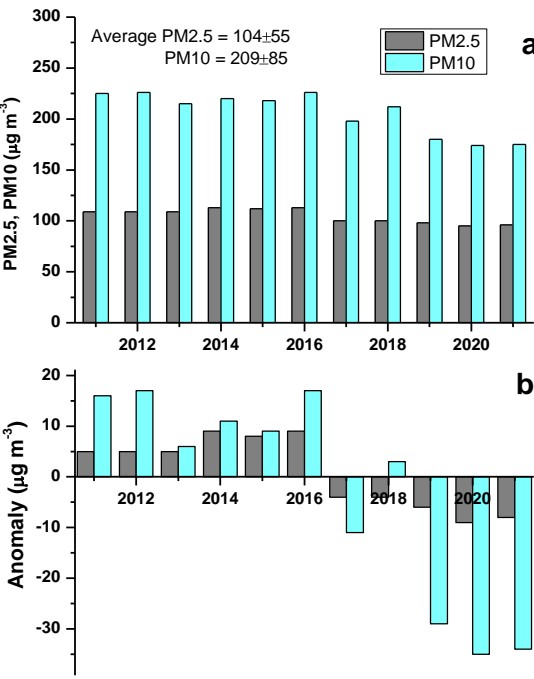

**Fig. 2 (a) Annual variation of PM$_{2.5}$ and PM$_{10}$ mass concentration with standard deviation in Delhi for 2011-2021, (b)**
**Anomaly of PM2.5 and PM10 for each year from the average (2011-2021) concentration**
Fig. 3 shows the seasonal variation of PM$_{2.5}$ and PM$_{10}$ from 2011 to 2021 in Delhi. It is detected that the highest
seasonal loading of PM$_{10}$ is during the post-monsoon (ON) and the lowest during monsoon (JJAS). Generally,
throughout the study period (2011-2021), average PM$_{10}$ loading over Delhi is noticed to be the highest in post-
monsoon (298±71), followed by winter (257±53), then pre-monsoon (207±52) and monsoon (127±50 µg/m³)
(Fig. 3(b)). PM$_{2.5}$ also showed a similar seasonal variation as PM$_{10}$ ((Fig. 3(a)). The average PM$_{2.5}$ was highest in
post-monsoon (170±50), followed by winter (145±39), pre-monsoon (83±22) and monsoon (59±19 µg/m³).



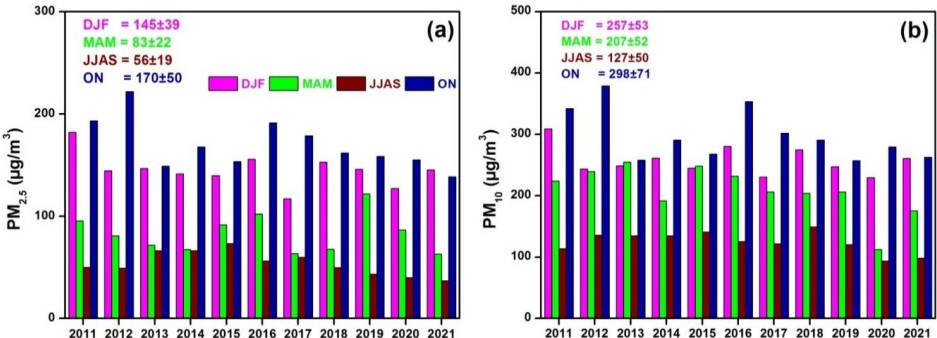

**Fig. 3 Seasonal variation of PM$_{2.5}$ and PM$_{10}$ (2011-2021) in Delhi**
**3.3 Linear trends of PM$_{2.5}$ and PM$_{10}$**
Fig. 4 and Table 1 show absolute annual and seasonal trends of PM$_{2.5}$ in Delhi during 2011-2021. A significant
declining (negative) trend is observed for PM$_{2.5}$ in Delhi with a definite change of -2.98 ± 0.53 μg/m$^3$ per year
(2.64 % reduction per year) or an overall 29.0 % reduction from 2011 to 2021 (Table 1). Singh et al., (2021)
reported a declining trend in five metro cities in India using US embassy data in each city, while Sharma et al.,
(2022) based on similar hourly data, concluded that no significant trend was witnessed. Hammer et al., (2020),
in their AOD-based global study, deciphered an increasing trend till 2012 for India and East Asia while Europe
and Eastern US showed a slow but steady reduction; however, the study further concluded that a global decline
in PM$_{2.5}$ is observed during 2011-2018 with India leading the pack with -0.54±0.7 μg/m$^3$/y. The recent works of
(Verma & Nagendra, (2022) and Chetna et al. (2022) based on six stations in Delhi show a drop of ~-5.1
μg/m$^3$/y (2014-2019) and   -1.35 μg/m$^3$/y (2007-2021) respectively in PM$_{2.5}$ respectively. PM2.5 observed in our
study may be more representative of decadal variation with -2.98 ± 0.53 μg/m$^3$ per year as it also presents a
varied combination of stations and hence NCR as a whole, though with a low representation of north-west
Delhi.
Reduction in PM$_{2.5}$ is attributable to changes in emissions, seasonal external influencing factors (like dust storms
and biomass burning) and meteorology over the study region (Verma & Nagendra, 2022; Chetna et al., 2022).
Chetna et al. (2022) detail the meteorological influences based on the re-analysis data to conclude that RH and
surface pressure increased temporally and wind speed decreased. While an increase in RH may help in the
deposition of particulates and low wind speed during summer may limit dust rising, a low wind would also help
build up concentration in winter due to lack of dispersion. Central and Delhi governments are implementing
various policies to curb air pollution in Delhi. Verma & Nagendra, (2022) provide a detailed timeline of such
policies. Our results also support the positive impact of such policies and some meteorological influences in the
declining trend revealed here.
Further, for meticulous analysis, linear trends are also calculated for different meteorological seasons in Delhi,
i.e. winter, pre-monsoon, monsoon and post-monsoon. Significant decreasing seasonal trends have been
observed for PM$_{2.5}$ during various seasons except in winter, where the trend in PM$_{2.5}$ is insignificant (P=0.085).
PM$_{2.5}$ has exhibited a declining trend of -4.51 ± 1.59, -3.43 ± 1.02, -2.35 ± 0.67 and -2.28 ± 1.28 μg/m$^3$/y



respectively, during post-monsoon, pre-monsoon, monsoon and winter. In their long-term seasonal trend study,
Chetna et al. (2022) found that winter displayed the slightest change with +0.06 $\mu g/m^3/y$, while the summer
showed the steepest reduction with -3.5 $\mu g/m^3/yr$. They also find a declining trend of -1.95 $\mu g/m^3/y$ in monsoon,
while current results indicate a stronger downward trend. The difference is attributable to different periods and
the number of observation stations. Our winter months include December, January and February, while their
study considering the latter two months resulted only in a slight incremental tendency for winter, unlike this one.
Similarly, the differences in post-monsoon trends are also due to the included month and other reasons.

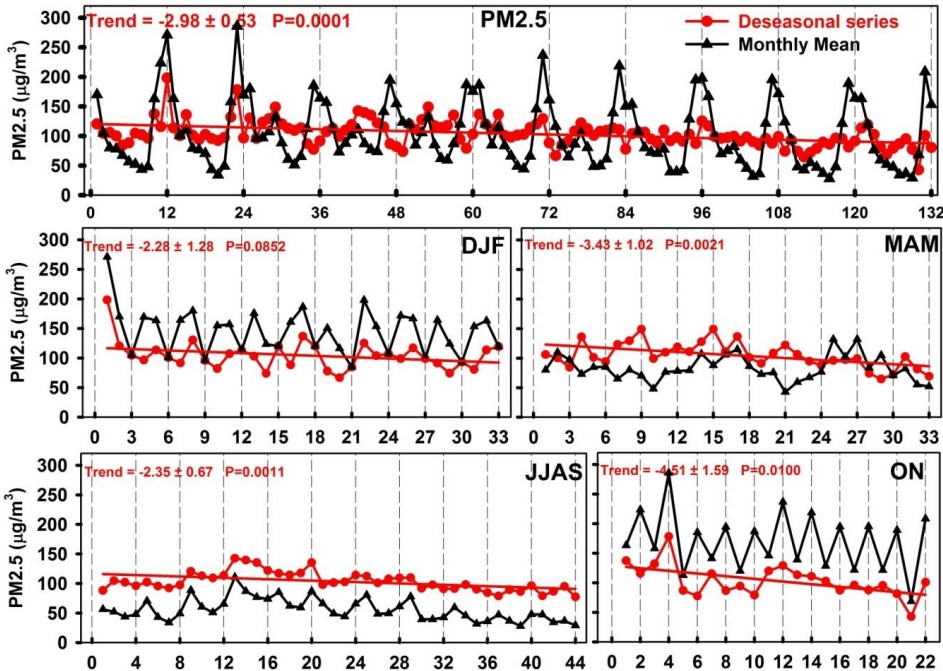

**Fig. 4 Time series of monthly averaged PM2.5 (black) deseasonalized series (red) with their corresponding linear fit**
**(with slope ± standard error) (red) for 2011-2021 in Delhi. Lower panels (4) consider DJF as winter, MAM,**
**Summer, JJAS, Monsoon; and ON, as post-monsoon.**
Absolute annual and seasonal trends of $PM_{10}$ in Delhi during 2011-2021 is shown in Fig. 5. Similar to $PM_{2.5}$, a
significant declining (negative) trend was noticed for $PM_{10}$ in Delhi with an absolute change of 4.91 ± 1.01
$\mu g/m^3$ per year (2.15 % reduction per year) or an overall 23.7 % reduction from 2011 to 2021 (Table 1).
Significantly decreasing seasonal trends have also been observed for $PM_{10}$ during various seasons except in
winter, where the trend in $PM_{10}$ was insignificant (P=0.2146). Seasonal $PM_{10}$ has decreased by 8.25 ± 2.29, 6.90
± 2.27, 3.08 ± 1.46 and 2.55 ± 2.01 $\mu g/m^3/y$ during pre-monsoon, post-monsoon, monsoon and winter,
respectively. The more significant decrease (w.r.t. 2011) in $PM_{10}$ was found during the pre-monsoon season and
was estimated as 3.66% followed by 2.57% decrease during the post-monsoon season.





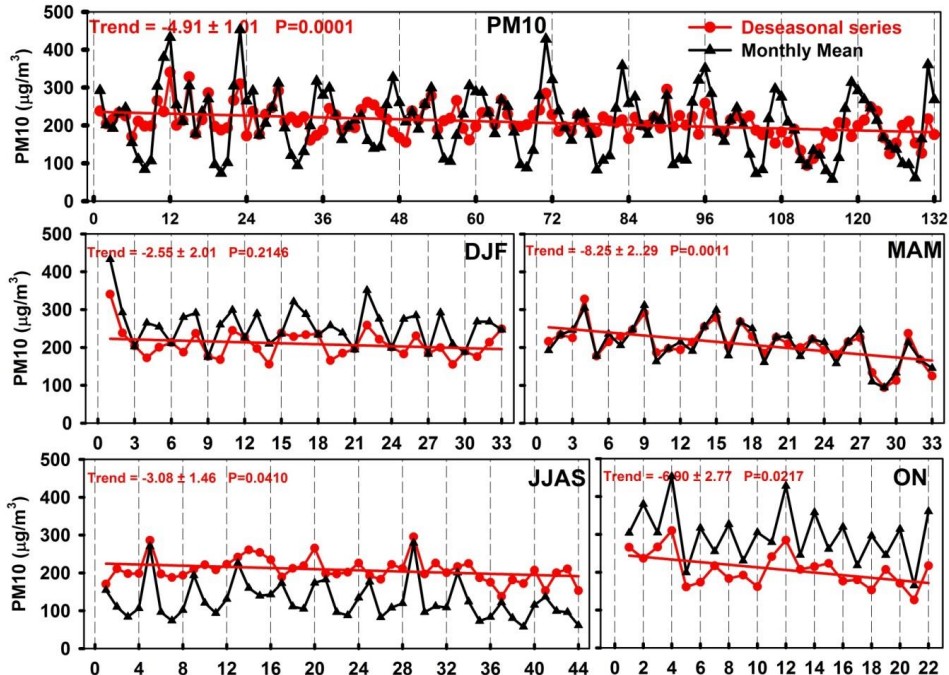

**Fig. 5 Same as in Fig.4 but for PM$_{10}$**
Table 1:  Reduction in mass concentration (slope ± standard error) and percentage reduction  per year) of PM$_{2.5}$
and PM$_{10}$ in Delhi for 2011-2021; P-values at 95% confidence level. The base year considered is 2011; annual
and seasonal means are tabulated.

|  | PM$_{2.5}$ (per year) | | | PM$_{10}$ (per year) | | |
|---|---|---|---|---|---|---|
|  | Trend (µg/m$^3$) | Relative Trend (%) | P-Value | Trend (µg/m$^3$) | Relative Trend (%) | P-Value |
| Winter | -2.28±1.28 | -1.62 | 0.0852 | -2.55±2.01 | -0.98 | 0.2146 |
| Pre-Monsoon | -3.43±1.02 | -3.55 | 0.0021 | -8.25±2.29 | -3.66 | 0.0011 |
| Monsoon | -2.35±0.67 | -2.40 | 0.0011 | -3.08±1.46 | -1.59 | 0.0410 |
| Post-Monsoon | -4.51±1.59 | -3.57 | 0.0100 | -6.90±2.77 | -2.57 | 0.0210 |
| Annual | **-2.98±0.53** | **-2.64** | 0.0001 | **-4.91±1.01** | **-2.15** | 0.0001 |

The most significant decrease has been observed during the post-monsoon season (3.57%), which may be
attributed partly to the change in meteorology and any trend in stubble-burning transport during this season. To
delineate the net effect of meteorology on PM concentration, a sensitivity study through WRF-Chem model
simulation of PM$_{2.5}$ is done for October in the post-monsoon season, as the season has shown the highest
negative trend compared to other seasons. The conclusions drawn from these simulations are systematically
presented further.
**3.4 Influence of meteorology on PM concentration**





To assess the impact of meteorology on PM concentration, WRF-Chem model sensitivity simulations have been
performed as discussed in section 2.3. According to the results, the weather conditions in the Delhi region in
2011 and 2015 were relatively more unfavourable, leading to higher levels of PM pollution than the weather
conditions in 2018 (Fig. 6). The adverse weather conditions in 2011 and 2015 resulted in an increase of 9.8%
and 5.1%, respectively, in meteorology-associated $PM_{2.5}$ with reference to that in 2018 (Fig. 6). Model results
also showed that unfavourable weather conditions contributed to an increase of 19.5% in meteorology-
associated $PM_{10}$ in 2011 and an increase of 11.7% in 2015 with reference to that in 2018 (Fig. 6). Thus changes
in meteorological conditions played a significant role in the long-term trends of $PM_{2.5}$ and $PM_{10}$ (Hammer et al.,
2021; Du et al., 2022; Chen et al., 2023). Gong et al.,(2021) estimated that the contribution of meteorology to
PM variation was 5% on an annual scale, whereas it escalated by 10-20 % during heavy pollution season in
China during 2013-2019. The meteorology-driven anomalies contributed −3.9% to 2.8% of the annual mean
$PM_{2.5}$ concentrations in eastern China (Xiao et al., 2021). Though there are independent studies of long-term PM
trends and meteorological variables, no studies have yet quantified the effect of meteorology on PM trends. Our
results indicate that the favourable meteorological conditions in 2018, compared to that in 2011 and 2015, are
instrumental in bringing down the PM levels by about 10%, at least for October. However, further studies are
required to quantify the approximate effect of each parameter. In the following section trends of some of the
common influences are considered.

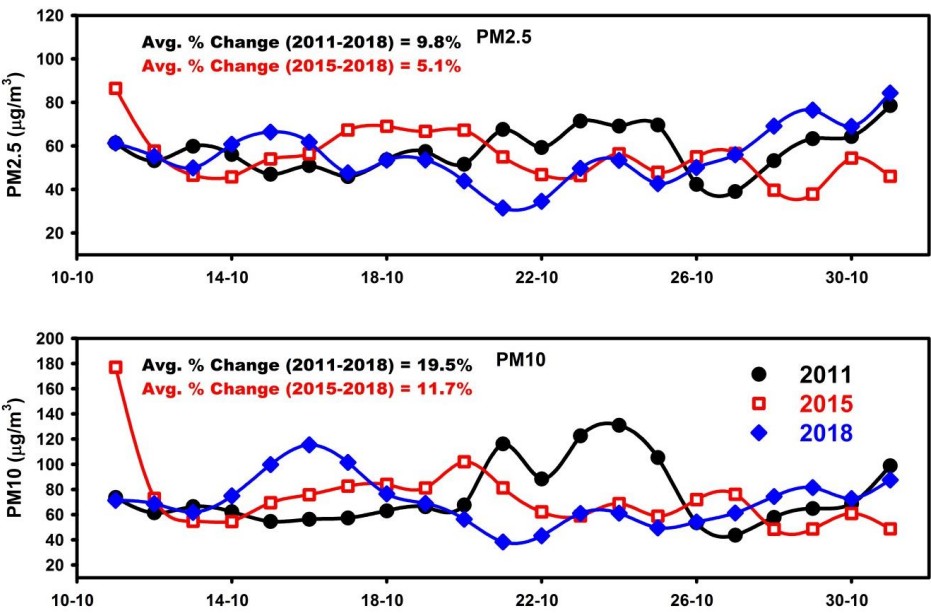

**Fig. 6 Simulated surface $PM_{2.5}$ and $PM_{10}$ over Delhi during post-monsoon due to change in meteorological conditions**
**of October: overall percentage change in the simulated $PM_{2.5}$ and $PM_{10}$ using meteorology of 2011 and 2015 with**
**reference to that of 2018.**



**3.5 Dust storms**
During the pre-monsoon season, dust storms impact Delhi air, resulting in high dust/coarse particulate
concentrations in $PM_{10}$ and, to a lesser extent, $PM_{2.5}$. Sarkar et al., (2019) deliberated upon the characteristics of
dust storm 2018 influence on the air of Delhi and adjacent areas. As explained earlier, the trend in dust storms is
calculated as they are potentially contributing external factors. It showed a decrease of 0.35 events per year
(Fig.7); hence, it may be said that overall, there is a reduction of 4 dust storms from 2011 to 2021. As evident in
Fig 7, some years have a very high impact, whereas some have a negligible impact; however, one cannot ignore
the temporal influence. This decrease in dust storms might have contributed a little, but its contribution to the
trend in $PM_{10}$ or $PM_{2.5}$ cannot be quantified as it is pretty complex.

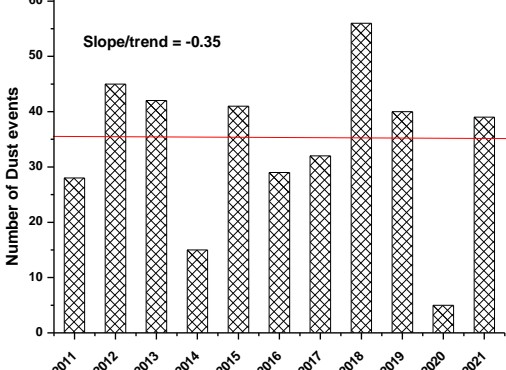

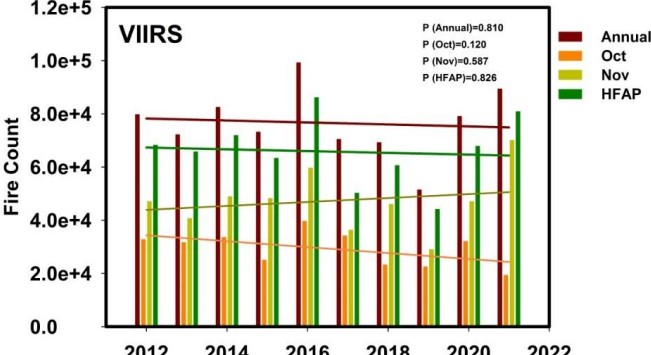

**Fig. 7 Trends in Dust storms (upper panel) and VIIRS Fire counts/stubble burning (lower panel) during 2011-2021**



**3.6 Stubble burning**
The surrounding region of Delhi has two significant stubble/crop burning periods: one in April-May and the
second in October-November. These events could also potentially impact Delhi's particulate matter (PM)
concentration. Nonetheless, the pre-monsoon burnings generally have lesser influence as the upwind direction
(southeast) during the period does not aid transport to Delhi, and mixed layer depth being high enough disperses
the transported pollutant efficiently. Conversely, during post-monsoon, the prominent upwind direction
(northwest) majorly aids PM transport from Haryana and Punjab. A study by Beig et al. (2020b) concluded that
air quality in Delhi during the post-monsoon season (October-November) was significantly influenced by
biomass burning/stubble burning exacerbated by prevailing winter conditions (Beig et al., 2020). Therefore, it
could be one of the seasonal external factors influencing the PM2.5 trend during 2011-2021. The annual trend in
satellite-derived (VIIRS) fire counts (lat: 27.67-33.42, long: 73.87-77.12), covering Haryana and Punjab, some
parts of northeast Rajasthan, and southwest Himachal Pradesh), a proxy to the intensity of stubble burning, was
estimated and found to be decreasing but negligible trend (Fig 7). Figure 7b also portrays the annual HFAP
(High fire activity period count) and October and November fire counts separately; though their trends are
different, all of them are statistically insignificant. The meaning of HFAP, including references, is given in the
supplementary file (S1). Fire count has a decreasing trend for October, whereas it has an upward tendency in
November. This decrease in stubble burning could account for only a tiny percentage of the declining $PM_{2.5}$
trend. As the reduction during post-monsoon is the highest, delving further, it is observed through the
concentration-weighted trajectories (Fig.8a) that the influence of crop burning in Haryana is more prominent in
this period. Quantifying the burning in terms of Fire Radiative Power (FRP), which is an average of fire
emission, the FRP trend over Haryana displayed a declining trend (Fig.8b), especially after 2016. This
observation supports the steeper reduction in the post-monsoon season. Again, the trend slopes are small and
insignificant for 2011-2021. This exercise is taken up only to demarcate the possible control region of PM
transport to Delhi during post-monsoon period.

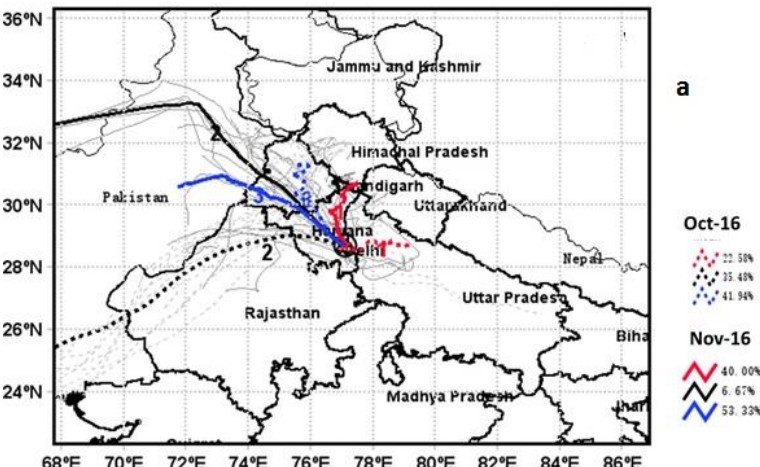

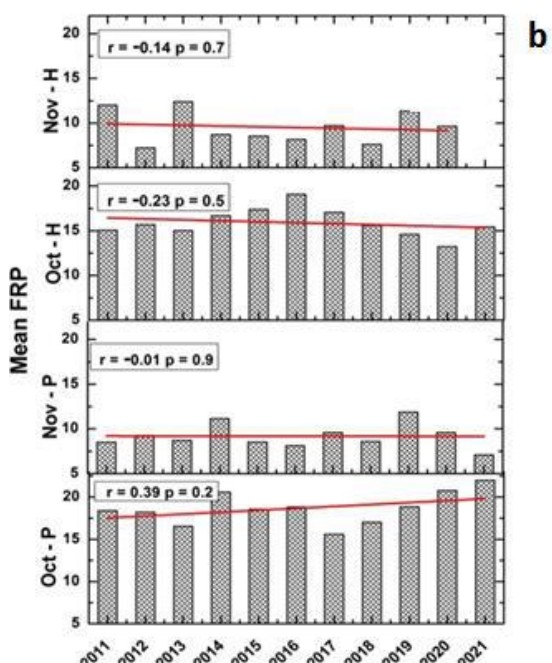

**Fig. 8 a) Concentration-weighted trajectories during October and November for a typical year and b) trends in FRP for Haryana (H) and Punjab (P) for the same period.**

Another critical factor that could affect the $PM_{10}/PM_{2.5}$ trend is the number of annual rainy days from 2011-2021. India Meteorological Department (IMD) gridded rainfall (25 km X 25 km) data was used to estimate the trend in rainy days (Pai et al., 2014). A rainy day is defined as a day with rainfall $\geq 2.5$ mm. The trend in annual rainy days was found to have negligible contribution to the PM2.5 trend, with a decrease of 0.06 rainy days in a year and an overall 0.7 rainy days during 2011-2021 (Fig.9). Chetna et al. (2022) found an increasing trend in humidity over Delhi through the wet deposition of PM or humidity-assisted growth and induced deposition is not always linear but complex.




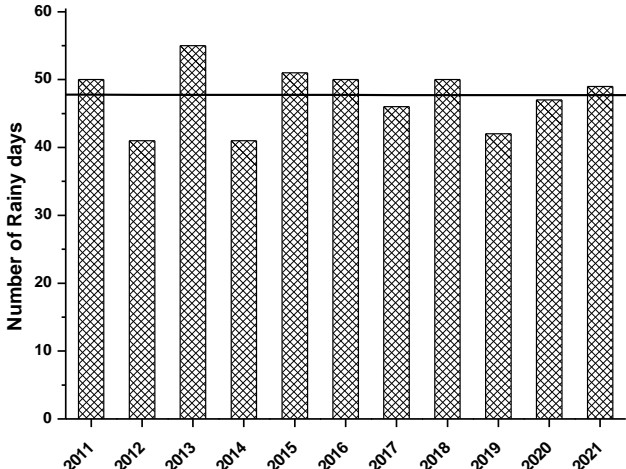

**Fig. 9 Annual number of rainy days during 2011-2021**
**3.7 Governmental Control Measures**
Despite accounting for meteorology and the three potential seasonal external factors, namely dust storms,
stubble burning and annual rainy days, one could not wholly explain the significant decrease in particulate
concentrations in Delhi during 2011-2021. Therefore, the improvement in air quality observed during 2011-
2021 could be attributable to various mitigation measures implemented in Delhi to curb air pollution.
In 2016, the vehicular density was ~8000 per 1000 population, which increased to 1.5 times in 2019 (Verma and
Nagendra, 2023). The BS IV norms were mandated in 2010 for personal cars and BS-VI in 2020, while BS-VI
for two/three wheelers were implemented in 2016. These were the traffic-related regulations during the study
period. A change from BS-III to BS-IV for 2 and 3-wheelers could have reduced the emission of PM by 50%
from that source. In 2015, NGT (National Green Tribunal) imposed a ban on diesel vehicles that are more than
10 years old.
On the industrial front, several restrictions have been continuously imposed on various facets such as cleaner
fuel, emission standards, stack height, etc., such as the pet coke and furnace oil ban in 2017, converting
industries to use CNG since 2018, creating new/stricter norms for emission reduction in various industries 2011
onwards and so on. On the societal front, improving public transport with cleaner fuel, increasing green cover
and specific initiatives to introduce cleaner cooking fuel, use of increased solar energy, shut down of power
plants and construction activities during adverse meteorological conditions also might have limited the PM
pollution. Specific studies of future scenarios show that a significant reduction in PM pollution is achievable by
stricter adherence to emission norms (Bhanarkar et al., 2018; Venkataraman et al., 2018; Conibear et al., 2018;
Chowdhury et al., 2019; Purohit et al., 2019). These comprehensive multi-pronged mitigation measures should



be able to explain about 15% reduction seen in particulate concentration trends other than meteorology and
external factors, especially since 2015.
Change in LULC over the years is one of the potential factors that can impact air quality in Delhi. Gupta (2021)
reported percentage change in LULC in national capital region Delhi as derived from high resolution satellite
imagery using geo-informatics. It is reported that 'Built-up Land area' changed by 5.46%, 'Agricultural land
area' by -4.95%, 'Forest area' by 2.91%, 'Barren & Scrub Land' by -3.18% and Water bodies by -0.24% during
2008-2018. Apparently, increase in Built-up and increase in forest cover was set off with decrease in
Agricultural area, Barren land and Water bodies owing to the pressures of population increase and for enacting
policy measures. Definitely, the increase of built up area at least at some point of time have contributed to
construction related dust but thereafter how they contribute to pollution cannot be assessed exactly as the net
effect of urbanization and open area dust rising contribution can be contradicting. While Barren land turning to
urban forestry is sure to reduce pollution except for pollen transport, if any. Similar is the case of Agriculture
area turning to Built-up as there may not be cropping throughout the year for non-availability of water and open
uncultivated land may be a source of dust in summer. Reduction of water bodies surely contribute to pollution in
any form. Overall observed LULC change during 2008-2018 seems not to play a decisive role in air quality
improvement because of the opposing outcomes.
**4. Conclusions**
A decade of in-situ SAFAR observations in Delhi has revealed gradual air quality improvement, specifically
from 2015 onwards. The linear trend analysis indicated that the observed $PM_{2.5}$ and $PM_{10}$ decreased 29% and
24%, respectively, from 2011 to 2021. Trends of seasonal external factors like dust storms, crop residue/stubble
burning, change in LULC over the years and the number of rainy days seemingly only insignificantly contribute
to the declining trend of particulate matter during 2011-2021. Sensitivity analysis using WRF-Chem to quantify
the role of meteorology reveals that over the years (from 2011 to 2018 for October), meteorological conditions
have become more favourable, contributing about 10% to the observed decreasing trend in $PM_{2.5}$. The decrease
in $PM_{2.5}$ observed on an annual scale could be attributed to the activities adopted from time to time to reduce
emissions, primarily and to meteorology to a lesser scale. Various mitigation plans implemented by governing
bodies to curb air pollution have improved Delhi's air quality over the years, manifested in the increased
number of satisfactory days (25 days in 2015 to 100 days in 2018). The study also finds that the Haryana region
has definitive control over the transport of pollutants from stubble burning to Delhi. The study's prime
conclusion is that Governmental policies and their efficient implementation, public initiatives and outreach can
turn even the most polluted cities into sustainable ones despite intensified multi-factored urbanization and
increasing emissions. Such long-term observations and their analyses and model evaluations can categorize such
effects.
**5. Author contribution**
'Yesobu' contributed to performed model simulation, analysis and visualisation, wrote first draft of manuscript.
'Latha' corrected manuscript, supervised analysis, reviewed the ms. 'Aditi' curated data from SAFAR stations
and partook in formal analysis. 'Siddhartha' provided logistic support for data collection and maintenance of
instruments. 'Murthy' involved in project management, guidance and final revision of manuscript.



**6. Competing interests**

The authors declare that they have no conflict of interest.

**7. Data availability**

The raw data supporting the conclusions of this article will be made available by the authors on request without undue reservation.

**8. Acknowledgement**

The authors are thankful to Dr. Beig who was instrumental in establishing and Dr. Shailesh Nayak and Prof. B N Goswami for initiating the SAFAR network. MoES is gratefully acknowledged for facilitating and funding for the maintenance of the same. The authors also acknowledge the Indian Institute of Tropical Meteorology, Pune, for the infrastructural and administrative support.

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
