# Peer review of "Observed improvement in air quality in Delhi during 2011- # 2021: Impact of mitigation measures"

_EGUsphere, 2024_

## Author Comment (AC1)

**REPLY to RC1**

Reply to each comment is marked in 'purple' colour

- **RC1**: 'Comment on egusphere-2024-803', Anonymous Referee #2, 15 May 2024  reply

1. the paper averages out 10 different locations in Delhi and describes the particulate matter scenario and attributes the decrease to the measures taken by the government. It would be better if the authors chose the locations immediately impacted by these government measures, instead of averaging them out. The microenvironments are very different from each other, thus averaging them out wouldn't depict the true picture.

   We understand your concern about the potential variability in microenvironments across different locations. Our initial approach was to provide a broad overview of the overall impact of government measures on air quality by averaging data from multiple locations. This is because the government implemented measures citywide rather than targeting specific areas.

2. 2020 was in full lockdown for few months, followed by partial lockdowns. Similar statewide lockdowns were observed during 2021 as well. So including 2020 and 2021 will skew the results. the anomaly seen in 2020 and 2021 is substantial compared to the other years, thus it doesn't depict the true picture. Please include the following years 2022, and 2023 to see if the trend persists and to rule out the impact of lockdown.

   Four Lockdown (LD1 to LD4) phases (with a gradual reduction in restrictions from LD1 to LD4) with LD1, LD2, LD3 and LD4 correspond to March 25-April 14, April15-May 03, May 04–17, May 18–31, 2020 respectively. No lockdowns implemented in 2021 in Delhi to the best of our knowledge. Some restrictions (event specific mitigation plans) are usually implemented every year whenever air quality reaches very poor condition.

   In order to understand whether the trend during 2011-2021 is affected by including the data with lockdown period during March 25-May31, 2020, we have calculated anomaly of each year from the decadal mean (2011-2021) and depicted in Figure 2b. The negative anomalies of PM2.5 in 2019, 2020 and 2021 are almost the same; hence inclusion/exclusion of 2020 data hardly changes the annual trend for the period, 2011-2021.

   The decreasing trend is observed from 2016 onwards as illustrated in Figure 2a in the manuscript.
* * *
   https://pib.gov.in/PressReleaseIframePage.aspx?PRID=1991970

   The above link leads to the 'Report of Ministry of Environment, Forest and Climate change', reporting a decreasing trend in PM10 and PM2.5 during 2018-2023 even after excluding 2020 values.
* * *
3. why was 2018 chosen as the meteorological base year for comparison?

.

The choice is based on certain facts. Delhi's 1st emission inventory was prepared in 2011. Revision of the same was done in 2018 and this is the same which was used in the current operational model. We could have taken 2016 but stopped short as we noticed that there is a kind of sharp drop in PM since 2016 while we analysed trend. Therefore, 2018 was chosen as a steady year to assess meteorological base. Further emissions data for this year are available in both the EDGAR and the local emission inventory SAFAR at the time of this manuscript preparation.

---

## Author Comment (AC2)

REPLY to RC2

Reply to each comment is marked in 'purple' colour

- **RC2**: 'Comment on egusphere-2024-803', Anonymous Referee #1, 15 May 2024  reply

  The manuscript evaluates the efficiency of air quality control policies in India. Notable reduction in particulate pollution was seen and the authors were able to estimate trends for the annual pollution levels. The topic of the manuscript is of interest for readers of ACP and overall, the presentation of the results is clear. However, some concerns need to be addressed before I can recommend the MS for publication.

  Thank you for your positive feedback on our manuscript and for highlighting its relevance to the readers of ACP. We appreciate your recognition of the clarity in our presentation and the significance of our findings on the efficiency of air quality control policies in India, particularly in the capital city, Delhi. We acknowledge your concerns and are committed to addressing them thoroughly to enhance the quality and consistency of our study. We are currently reviewing your specific comments and will make the necessary revisions to ensure our manuscript meets the high standards of ACP.

  My main concern is that the method for trend estimation is not statistically sound, and it is not capable of answering the questions researchers are trying to solve. The 13-month moving average is claimed to de-seasonalize the data, but it only smooths the variation. The trend probably is not linear and t-test is definitely not a method for calculating a trend. With appropriate trend fitting methods, deseasonalization is not even needed but the seasonal variation can be taken account in the trend calculation.

  Thank you for your concern regarding the trend calculation. We have followed the methodology used by various researchers, as referenced in Section 2.4 of the manuscript.

  Page 6, Section 2.4, Lines 24-31: 'For the trend calculation over Delhi for 2011-2021, we used monthly averaged concentrations of PM2.5 and PM10. The datasets were first de-seasonalized by applying a 13-month moving average to obtain an initial trend estimate, followed by a stable seasonal filter to remove the seasonal cycle. Linear regression was then applied to the de-seasonalized time series of PM2.5 and PM10 to calculate the linear trend. The statistical significance of the linear trend was evaluated using a parametric student t-test, and only statistically significant non-zero slopes (p-value < 0.05) were presented.'

  **Specific comments:**

  Page 7, lines 24-25: averaging does not eradicate inhomogeneity. By averaging, the researchers just assume data "homogenic enough" to get representative city-level value. How justifiable this assumption is proposes another question. I would suggest a sensitivity analysis (perhaps shown in the supplement) where basic statistics would be shown and appropriate trends would be fitted to individual datasets.

  The sources are heterogeneous across Delhi city; hence AQMS are located representing different microenvironments (like commercial, residential, industrial, etc) so that overall air quality of Delhi can be estimated by taking average of all stations following WMO guidelines. We will estimate each station data statistics to understand characteristic features as suggested separately.

Page 9, line 28: Announcing p=0.085 as insignificant is a bit of overkill. Interpretation for p-value should not be based on some artificial threshold value but it should be treated as quantitative measure of significance. See e.g. https://doi.org/10.1080/00031305.2016.1154108 and https://www.nature.com/articles/d41586-019-00857-9

Thank you for your comment. We appreciate your suggestion regarding the interpretation of p-values. We agree that p-values should be treated as a quantitative measure of significance rather than being subjected to an artificial threshold. As suggested, we will revise the manuscript to reflect this perspective. Specifically, we will interpret the p-value of 0.085 as indicating a moderate significance level and discuss its implications within the context of our study. Additionally, we will reference the guidelines provided in the cited articles to support our revised interpretation.

Section 3.4. The argument on the effect of meteorology on PM needs confirmation. The comparison of model results in different time points does not quantify the effect. This could be done with the data by using multivariable statistical models like applied here https://doi.org/10.5194/acp-20-12247-2020 or advanced time series methodology introduced here http://urn.fi/URN:NBN:fi:jyu-201603111829 and here http://dx.doi.org/10.1007/978-3-030-21718-1_4. The same methods can also be applied in Section 3.5. in quantification of the dust storms and in 3.6. to account for stubble burning.

Thank you for your valuable comment. We appreciate the suggestion to strengthen our argument on the effect of meteorology on PM by using more rigorous quantitative methods. We followed an approach by various researchers found in the literature (Zhang et al., 2021; Hammer et al., 2021; Du et al., 2022)

Du, H., Li, J., Wang, Z., Chen, X., Yang, W., Sun, Y., Xin, J., Pan, X., Wang, W., Ye, Q., & Dao, X. (2022). Assessment of the effect of meteorological and emission variations on winter PM2.5 over the North China Plain in the three-year action plan against air pollution in 2018–2020. Atmospheric Research, 3 280(August), 106395. https://doi.org/10.1016/j.atmosres.2022.106395

Hammer, M. S., Donkelaar, A. Van, Martin, R. V., McDuffie, E. E., Lyapustin, A., Sayer, A. M., Hsu, N. C., 25 Levy, R. C., Garay, M. J., Kalashnikova, O. V., & Kahn, R. A. (2021). Effects of COVID-19 lockdowns on fine particulate matter concentrations. Science Advances, 7(26), 1–11. https://doi.org/10.1126/sciadv.abg7670

Zhang, Y., Ma, Z., Gao, Y., & Zhang, M. (2021). Impacts of the meteorological condition versus emissions reduction on the PM2.5 concentration over Beijing–Tianjin–Hebei during the COVID-19 lockdown. Atmospheric and Oceanic Science Letters, 14(4), 100014. https://doi.org/10.1016/j.aosl.2020.100014

https://doi.org/10.5194/acp-20-12247-2020; http://urn.fi/URN:NBN:fi:jyu-201603111829 and http://dx.doi.org/10.1007/978-3-030-21718-1_4) are good pointers to be used in the future work, we appreciate your suggestion. However, for this study we are short of station wise meteorological variables for the study period.